# CCGAN as a Tool for Satellite-Derived Chlorophyll *a* Concentration Gap Reconstruction

Leon Ćatipović [1],*, Frano Matić [2], Hrvoje Kalinić [1], Shubha Sathyendranath [3], Tomislav Županović [1], James Dingle [3] and Thomas Jackson [3]

1   Environmental Data Analysis Laboratory, Faculty of Science, University of Split, 21000 Split, Croatia
2   University Department of Marine Studies, University of Split, 21000 Split, Croatia
3   National Centre for Earth Observations, Plymouth Marine Laboratory, Plymouth PL1 3DH, UK
*   Correspondence: leon.catipovic@pmfst.hr

**Abstract:** This work represents a modification of the Context Conditional Generative Adversarial Network as a novel implementation of a non-linear gap reconstruction approach of missing satellite-derived chlorophyll *a* concentration data. By adjusting the loss functions of the network to focus on the structural credibility of the reconstruction, high numerical and structural reconstruction accuracies have been achieved in comparison to the original network architecture. The network also draws information from proxy data, sea surface temperature, and bathymetry, in this case, to improve the reconstruction quality. The implementation of this novel concept has been tested on the Adriatic Sea. The most accurate model reports an average error of 0.06 mg m$^{-3}$ and a relative error of 3.87%. A non-deterministic method for the gap-free training dataset creation is also devised, further expanding the possibility of combining other various oceanographic data to possibly improve the reconstruction efforts. This method, the first of its kind, has satisfied the accuracy requirements set by scientific communities and standards, thus proving its validity in the initial stages of conceptual utilisation.

**Keywords:** GAN; generative adversarial network; reconstruction; satellite chlorophyll *a*

## 1. Introduction

Satellite oceanography is a key tool in oceanographic measurements, but its usefulness is hindered by frequent gaps in data [1]. Solving the issue of missing data has been the aim of many solutions throughout the past two decades. During this time, a number of different reconstruction methods and approaches have been designed and implemented. The earliest attempts at gap-filling included data merging from multiple sensors [2], kriging [3], and basic regression techniques [4]. One of the most popular methods is the so-called Data Interpolating Empirical Orthogonal Functions (DINEOF) [5]—a linear reconstruction method based on empirical orthogonal functions. Lately, reconstruction techniques in oceanographical remote sensing have been turning toward the utilisation of various machine learning techniques. These techniques range from support vector machines [6], random forests [7] to various neural networks [8–10]. These techniques have seen an ever-increasing popularity, with an emphasis on neural-network-based approaches, which have seen an extreme surge in implementation, practicality and popularity in the past several years [11].

One special architecture of the neural networks, called the Generative Adversarial Network (GAN) represents the next leap in the neural-network reconstruction approach [12]. Unlike other neural networks, GANs represent a unity of two separate networks, a generator and a discriminator, "pitted against" each other. While the generator is tasked with generating new data, the discriminator is tasked with discerning whether or not the data presented to it originate from a real source or from the generator. This way, based on the feedback information, both networks become more proficient in their respective tasks, resulting in the generation of realistic data [12]. GANs have been successfully utilised

within the domain of satellite oceanography for reconstruction purposes [13–16], but their application has been limited to sea surface temperature (SST) only [11]. Outside of strictly reconstruction-oriented purposes, GANs have seen a vast range of utilisation in oceanography [17–20].

This paper aims to showcase the concept and validity of reconstructing oceanographic data using a variation of Generative Adversarial Network, so-called Context Conditional Generative Adversarial Network (CCGAN) [21]. While the generator of typical GAN usually transforms a random input to generate new data, CCGAN inputs corrupt data and only generates the missing part based on the available data, or the context, hence the name [21]. The original development of Context Conditional Generative Adversarial Network focused on natural image reconstruction only and proved quite successful [21]. However, this particular architecture has not yet been used in the reconstruction of oceanographic data. This work utilises an adjusted version of CCGAN for missing chlorophyll *a* data ($chl_a$) reconstruction. The paper also examines the effects of utilising proxy data—correlated data originating from an external source (e.g., different sensor)—on the reconstruction accuracy. In this case, chlorophyll *a* data are augmented by sea surface temperature data and bathymetry data, both of which are correlated to chlorophyll *a* [22,23]. Rather than deriving the complex numerical correlation between the three to obtain the absolute chlorophyll *a* concentration, sea surface temperature has been included to better derive water masses that ultimately dictate the fronts and shape of surface chlorophyll *a* concentration, while bathymetry serves as an indicator of distance from the shore in the Adriatic [24], helping to discern high chlorophyll *a* coastal areas from oligotrophic open waters.

To gauge the validity of this reconstruction method, the International Organization for Standardization (ISO) standard for in situ measurements [25] and Essential Climate Variable (ECV) product requirements [26] will be taken into account.

## 2. Data and Methods

Rather than selecting open ocean regions, which are generally homogeneous regarding surface $chl_a$, the Adriatic was chosen as the area of interest due to a considerable amount of literature documenting oceanographic processes, different water masses, complex bathymetric and coastal structure, effects of fluvial and atmospheric influence, and so on [24]. Wet points, meaning points representing the sea in the area encapsulated by 40–46° N and 12–20° E, were considered but the Tyrrhenian Sea was discarded. The area of study is depicted in Figure 1.

### 2.1. Data Sources

Ocean Colour-Climate Change Initiative (OC-CCI) Version 5.0 Data suite [27] was created by band-shifting and bias-correcting the data from Sea-viewing Wide Field-of-view Sensor, Moderate Resolution Imaging Spectroradiometer, Visible Infrared Imaging Radiometer Suite, and Ocean and Land Colour Instrument data to match the Medium Resolution Imaging Spectrometer data [27]. The chlorophyll *a* concentration was derived using an array of algorithms [28] based on the target water class membership [29]. With a spatial resolution of 1 km, this resulted in matrix dimensions of $576 \times 768$. Time frame spanned from 1 January 2003 to 31 December 2020, a total of 6575 days. All data are publicly available and accessible at www.oceancolour.org accessed on 1 August 2023.

To determine whether a machine learning model for reconstructing missing $chl_a$ could be improved, additional variables have been included. These variables are commonly referred to as proxy variables since they are used as an instrument for chlorophyll reconstruction. In this study, two additional proxy variables were considered: SST and bathymetry. Daily satellite SST was retrieved from Group for High Resolution Sea Surface Temperature Level 4 dataset (available at: www.podaac.jpl.nasa.gov/dataset/MUR-JPL-L4-GLOB-v4.1 accessed on 1 August 2023) [30]. As the SST data resolution was higher than the $chl_a$ resolution, it was scaled down to match the $576 \times 768$ grid. SST data were time-matched with $chl_a$. Bathymetrical data are time-independent, therefore only one instance was obtained

from the General Bathymetric Chart of the Oceans (available at www.gebco.net accessed on 1 August 2023) [31]. Bathymetry resolution was scaled identically as SST, and the data were subjected to cutoffs at −1000 and 400 m. The cutoff at 400 m was imposed to differentiate between low-lying coastal areas and mountainous coastal areas as coastal regions above the cutoff should not have a significantly different effect on the chl$_a$ regardless of the altitude, if such effects should even take place. Similarly, the seafloor at the maximum depth of the Adriatic at 1233 m would probably not affect the chl$_a$ significantly differently than a seafloor at 1000 m depth.

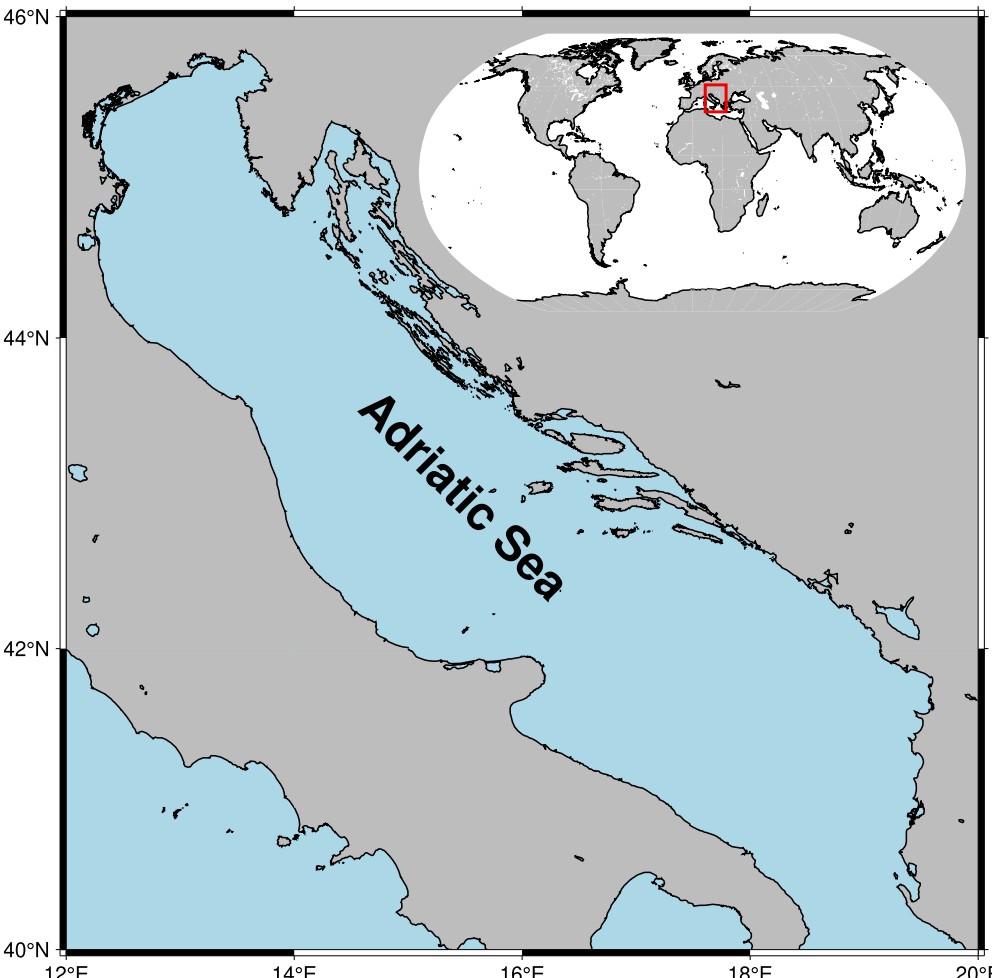

**Figure 1.** The Adriatic Sea. The position on the globe is enclosed by the red rectangle on the minimap.

*2.2. Dataset Integration*

The bane of any machine learning algorithm is the sheer amount of representative data required for training [32]. This becomes an even bigger issue when the original data have gaps, which is certainly the case for satellite-derived chl$_a$. Since there is no go-to method for creating a gap-free and representative dataset, a heuristic approach was developed. As the input of the neural network is three $64 \times 64$ data matrices, the dataset creation process implies sampling $64 \times 64$ subsets from the original $576 \times 768$ locations area from day to day while being subject to certain rules. The details and outcomes of the dataset creation method have been explored in the Appendices A and B. The three matrices that go into the neural network contain chl$_a$, SST and bathymetrical data, respectively. The matrices are stacked vertically so that the final shape of the input is $3 \times 64 \times 64$. Because SST and bathymetrical data are gap-free, the sampling is therefore solely determined by the chl$_a$ data.

### 2.3. Context Conditional Generative Adversarial Network

CCGAN is a variation of the established GAN architecture. The GAN architecture is based on pitting two separate networks against each other: the generator network and the discriminator network [12]. Simplified, the purpose of the generator is to *generate* a distribution $p_g$ over the data $x$ so that it matches the original data distribution $p_x$ as closely as possible. The output of the generator for some input data $z$ from distribution $p_z$ is defined as $G(z)$. On the other hand, the discriminator *discriminates* whether or not its input originated from $p_x$ or $p_g$, by outputting a probability value $D(G(z))$. The aim of the discriminator is to be as accurate as possible when discriminating between the real and fake input. GAN functions by simultaneously training both the generator and discriminator in a min-max game with a value function $V(G, D)$, also known as *loss* function, given as [12]:

$$\min_G \max_D V(D, G) = \mathbb{E}_{x \sim p_x}[\log D(x)] + \mathbb{E}_{z \sim p_z}[\log(1 - D(G(z)))]. \tag{1}$$

While some implementations and variations of GAN take random noise $z$ as input [12,33], CCGAN takes masked/corrupted/missing data as input and fills in the missing parts based on the available context data around the missing parts [21]. This way, rather than transforming some random noise input $z$, CCGAN exploits available data to optimise the generation of missing data. Formally, let $m$ be a binary mask that will obscure some parts of the data. Then, the generator receives $m \odot x$, where $\odot$ denotes element-wise multiplication. The *loss* function from Equation (1) then becomes [21]:

$$\min_G \max_D V(D, G) = \mathbb{E}_{x \sim p_x}[\log D(x)] + \mathbb{E}_{x \sim p_x, m}[\log(1 - D(G(m \odot x)))], \tag{2}$$

With this formulation, the output of the generator becomes $x_G = G(m \odot x)$. However, this is not the final reconstruction result, as the generator is not focused on reconstructing data not obscured by the binary mask [21]. Fully reconstructed, or better yet, the inpainted matrix $x_I$ is obtained via [21]:

$$x_I = (1 - m) \odot x_G + m \odot x. \tag{3}$$

Additionally, CCGAN has an optional input of the complete/uncorrupted/unmasked matrix of a lower resolution $x_R$ which improves the reconstruction accuracy [21]. Low-resolution input has been obtained through bilinear interpolation to $16 \times 16$ points. Given the original size of the matrix was $64 \times 64$ points, the low-resolution matrix contained just 6.25% of the original information. Finally, with this additional input, the *loss* function 2 became:

$$\begin{aligned}\min_G \max_D V(D, G) &= \mathbb{E}_{x \sim p_x}[\log D(x)] \\ &+ \mathbb{E}_{x \sim p_x, x_R \sim p_x, m}[\log(1 - D(G(m \odot x, x_R)))],\end{aligned} \tag{4}$$

The architecture of the network remained mostly the same [21], except that the outputting layer of the generator was changed from a hyperbolic tangent function to a sigmoid function, for reasons explained in the following paragraphs. Generator [33] consists of six downscaling and six upscaling layers, based on 2D convolution and transposed-convolution operators, respectively. Both operators consisted of kernel size $4 \times 4$, a stride of $2 \times 2$, and a padding of 1. Discriminator was based on the VGG-A network [34] without the fully connected layers [21]. Adam optimiser [35] was utilised. Learning rate was set at 0.0002, momentum term was set at 0.5. The remainder of the hyperparameters were left unchanged [21,35].

Each model was trained for 50 epochs, and each training dataset was split into batches of 20 data matrices. The number of epochs and batch size were purposefully set identically for each set to minimise the algorithm effect and maximise the dataset variability effect on the reconstruction accuracy. During the training process, at every 50 batch passes, the generator and discriminator loss values were recorded for later analysis. For the sake of convenience and to showcase the robustness of the CCGAN, a square mask will be applied

to the middle of the training data matrix, so that a large and important portion of the data is obstructed. With the dimensions of the data matrix being $64 \times 64$ and the dimensions of the mask being $32 \times 32$, 25% of the data are covered, as depicted in Figure 2.

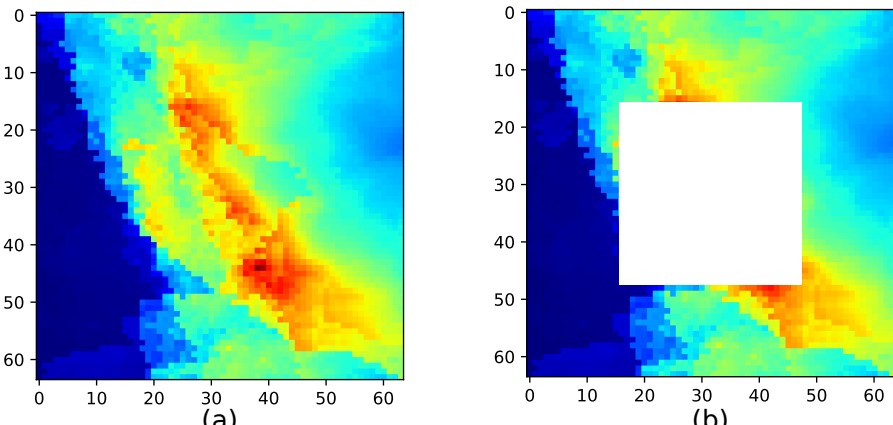

**Figure 2.** Visualisation of the masking process of chl$_a$ data (**a**) representing the full data; and (**b**) representing the manually masked data.

For this research, the *PyTorch* implementation of CCGAN was used [36,37]. The implementation emulates semi-supervised learning based on arbitrary labels used for defining real data and fake data. Labels are defined as appropriately shaped tensors T and F filled with ones and zeros, representing real and fake data, respectively. For practicality, the original loss function given by Equation (4) was split into generator loss:

$$L_G = \|D(G(m \odot x, x_R)), T\|_2^2 \tag{5}$$

and discriminator loss, given as:

$$L_D = 0.5 \cdot \left[ \|D(x), T\|_2^2 + \|D(G(m \odot x, x_R)), F\|_2^2 \right], \tag{6}$$

where $\| \ \|_2$ represents the mean squared error (squared L2 norm, MSE) between each element. While this implementation [36], denoted as the MSE$_1$-based model when used on CelebA Dataset [38], provided satisfactory results, if the loss functions defined in this way were to be used on chl$_a$ data, the discriminator would significantly outperform the generator during training, leading to poor reconstruction accuracy, as represented in Figure 3. To mitigate this problem, updated loss functions were proposed:

$$L_G = \|G(m \odot x_1, x_R) \odot (1 - m_{LC}), \ x_1 \odot (1 - m)\| \tag{7}$$

and

$$\begin{aligned} L_D = 0.5 \cdot [&\|D((1 - m) \odot x_1), T\|_2^2 \\ &+ \|D(G(m \odot x_1, x_R) \odot (1 - m_{LC})), F\|_2^2]. \end{aligned} \tag{8}$$

These loss functions are a part of the MSE$_2$-based model. Two changes are to be noticed here. Firstly, x$_1$ denotes the chl$_a$ data, the only data that are taken into account for training evaluation, as only chl$_a$ is targeted for reconstruction. Secondly, as the positions of land points, as defined by the dynamic land–sea mask, are always known for each data matrix, there is no need to gauge reconstruction accuracy on points that are known to be land points. Therefore, the land points from the real data were superimposed onto the respective generated data before being subjected to evaluation. This way, the training process will be streamlined and further optimised as only actual sea points will be subjected to testing. This change is implemented via the $m_{LC}$ mask. However, even this update to the loss functions resulted in poor reconstruction accuracy, mainly because, in this instance, the generator outperformed the discriminator, as depicted in Figure 3. A possible cause of

such poor results might lie in the MSE itself. While the MSE might keep numerical values in check, it cares little for the actual structural distribution of the data, as exemplified in Figure 4. Therefore, another model was proposed, whose generator loss function has been swapped for:

$$L_G = 1 - \psi[G(m \odot x_1, x_R) \odot (1 - m_{LC}), \ x_1 \odot (1 - m)] \tag{9}$$

The $\| \ \|_2$ function has been replaced with Structural Similarity Index function ($\psi$) [39,40]. $\psi$ is defined where $a$ and $b$ are square matrices, $\mu_a$ and $\mu_b$ are mean values of $a$ and $b$, $\sigma_a^2$ and $\sigma_b^2$ are variances of $a$ and $b$, $\sigma_{ab}$ is the covariance of $a$ and $b$, $c_1 = (0.01L)^2$ and $c_2 = (0.03L)^2$ are variables based on the dynamic range $L$ of data-values [39]. $\psi$ outputs a value in the range $[-1, 1]$, with 1 indicating a perfect copy, while $-1$ denotes maximal dissimilarity. Since $\psi$ measures the similarity rather than difference, the loss value in Equation (9) is defined as $1 - \psi$. As $\psi$ is undefined for negative values which can occur as an output of the final layer of the generator which contains a hyperbolic tangent function, the function was replaced with sigmoid function, ensuring all outputs are positive. $\psi$ was implemented using *piqa* package [41]. The motivation behind including $\psi$ is similar to the inclusion of SST: while numerical accuracy in reconstruction is important, the shape of the reconstructed features is no less important. These changes resulted in a stabler training process, as depicted in Figure 3, resulting in better reconstruction accuracy. This model was denoted as the Structural Similarity Index Measure (SSIM)-based model.

*2.4. Error Metrics*

To quantise the reconstruction accuracy, three error metrics were implemented. The definitive accuracy of reconstruction was measured only on the masked part. The value of MSE was calculated as:

$$\text{MSE} = \|G(m \odot x_1, x_R) \odot (1 - m_{LC}), \ x_1 \odot (1 - m)\|_2^2. \tag{10}$$

Secondly, SSIM was given by:

$$\text{SSIM} = \psi(G(m \odot x_1, x_R) \odot (1 - m_{LC}), \ x_1 \odot (1 - m)). \tag{11}$$

The final error metric was a relative error (RE) as defined by:

$$\text{RE} = \frac{1}{M} \sum_{i=1}^{M} \frac{|(G(m \odot x_1, x_R) \odot (1 - m_{LC}))_i - (x_1 \odot (1 - m))_i|}{(x_1 \odot (1 - m))_i} \times 100\% \tag{12}$$

This metric was added in order to put a more quantifiable and tacit measure of error. By dividing with the data values, RE can encounter division by zero if the land data happen to be subjected to evaluation. To avoid this, RE was calculated using only sea data denoted by $_i$. The total number of sea points per data matrix is $M$. While generally higher, SSIM also means a lower MSE and RE, measures which are not equivalent and may produce significantly different reconstruction results. While MSE and RE are relatively easily physically interpretable, SSIM does not share that feature. Therefore, it is natural to question the motivation behind the use of SSIM. However, it is easy to imagine a case where MSE and RE fail. As a showcase where only relying on numerical values could significantly hinder the structural realism of reconstruction, Figure 4 is presented. As can be noticed, structurally significantly different matrices are virtually equidistant from the original matrix in terms of MSE and RE, while this is not the case with the SSIM. If solely relying on MSE (or RE), it would be difficult to determine which matrix is a better representation of the original. Thus, when aiming at matrix reconstruction optimisation, SSIM better captures the structure of the data than either MSE or RE.

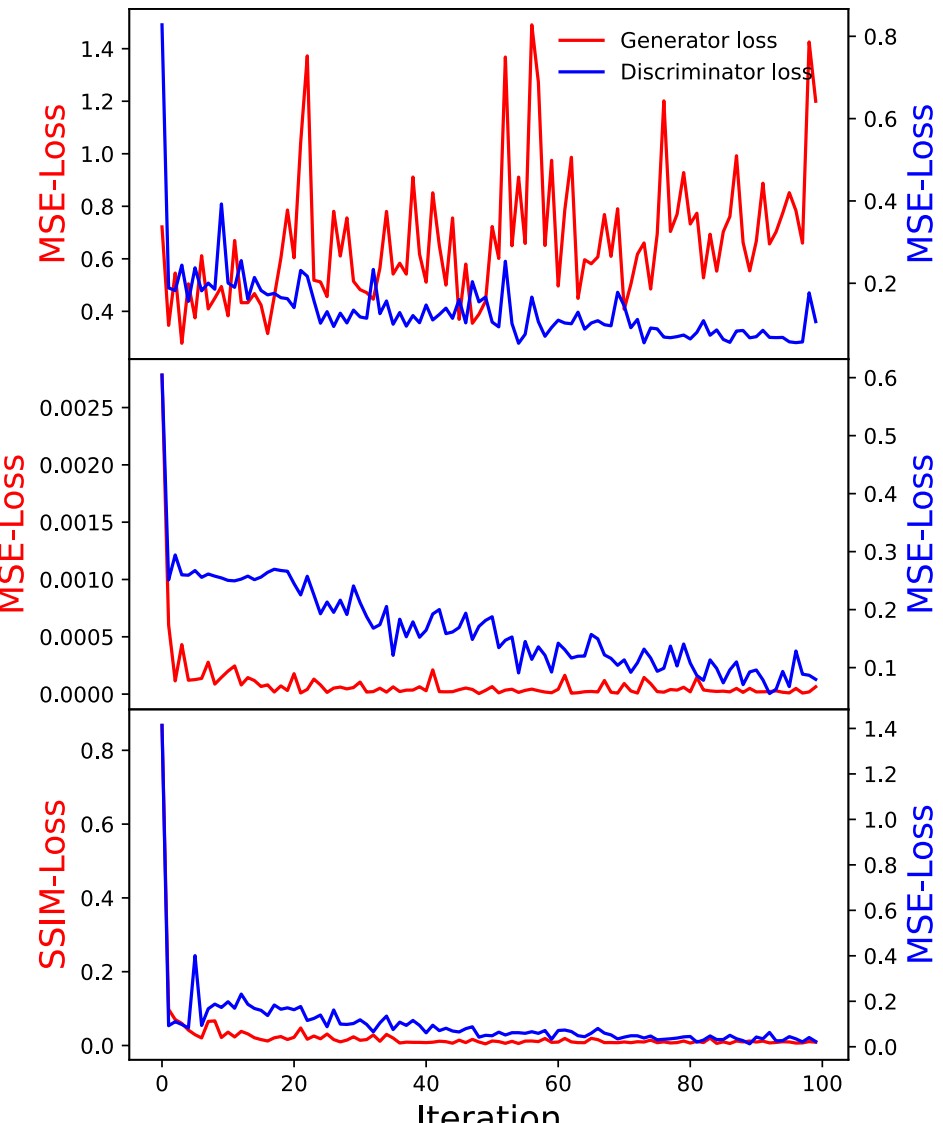

**Figure 3.** Iteration-dependent of the behaviours of generator and discriminator loss values based on two different loss metrics. The top graph displays the mean-squared-error-based (MSE) model. The middle graph depicts the behaviour of the MSE model that takes into account the distribution of land points. Bottom graph displays the Structural-Similarity-Index-Measure-based (SSIM) model.

### 2.5. Growing Neural Gas

To provide the most common examples of reconstruction, the Growing Neural Gas (GNG) [42] was implemented to extract the characteristic patterns from the testing dataset. Input into the algorithm consisted of the $3 \times 64 \times 64$ data matrix formatted into a data vector containing 12,288 features. Rather than implementing GNG once on the test dataset (containing over 50,000 of the aforementioned data vectors, see Appendices B and C) and obtaining a presentable number of patterns (e.g., around 10), GNG was implemented twice to avoid the oversmoothing of the data. First, the implementation reduced the dataset to 100 patterns. These 100 patterns were further fed into GNG in order to obtain the eight patterns used for the visualisation of reconstruction examples. Using these eight final patterns, also known as best matching units (BMUs), the closest possible dataset examples were determined using the least vector norm. The GNG algorithm itself was implemented in Python's library NeuPy using fixed parameters: step = 0.1, neighbour step = 0.001, maximum edge age = 50, number of iterations before adding a neuron = 100, aftersplit error decay rate = 0.5, error decay rate = 0.995, and minimum update distance = 0.2. Each run lasted 300 epochs.

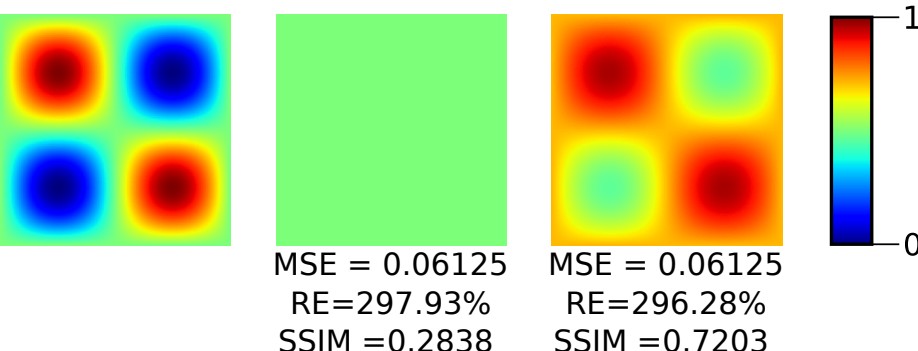

MSE = 0.06125    MSE = 0.06125
RE=297.93%     RE=296.28%
SSIM =0.2838    SSIM =0.7203

**Figure 4.** Example of MSE ambiguity. The leftmost image was created by the cross multiplication of a sinusoidal vector and its transpose. The middle image was filled with the mean value of the leftmost image. The rightmost image was obtained by the linear transformation of the leftmost image.

## 3. Results and Discussion

### 3.1. Verification of SSIM-Based Model

In order to justify changes to the architecture described in Section 2.3, a brief comparison between MSE-based and SSIM-based CCGAN models was made. Before calculating the metrics, data were rescaled to reflect the physical $chl_a$ values in mg m$^{-3}$. Results are displayed in Table 1. Not surprisingly, due to the training behaviour depicted in Figure 3, MSE-based models (depicted in the top and middle graphs) failed to optimise properly, and, even though the latter performed slightly better, both resulted in poor reconstruction accuracy. The SSIM-based model significantly outperformed both MSE-based models. Interestingly enough, even though the SSIM-based model is trained solely on the Structural Similarity Index function, MSE and RE metrics also show improvements. Figure 4 displays how the MSE-based models failed to optimise properly, causing significant noise in the training process.

**Table 1.** Comparison of the three reconstruction accuracy metrics for the three models with different loss functions. $MSE_1$ represents the base model whose loss function is determined solely by the mean squared error value; $MSE_2$ is the updated mean squared error model whose loss function takes into account the distribution of land points; and SSIM is the model whose loss function has exchanged the mean squared error metric for Structural Similarity Index Measure. $\mu$ represents the mean, and $\sigma$ represents the standard deviation.

| | $\mu_{SSIM}$ | $\sigma_{SSIM}$ | $\mu_{MSE}$ | $\sigma_{MSE}$ | $\mu_{RE}$ (%) | $\sigma_{RE}$ (%) |
|---|---|---|---|---|---|---|
| $MSE_1$-based model | 0.09 | 0.14 | 29.61 | 8.26 | 2570 | 1445 |
| $MSE_2$-based model | 0.12 | 0.15 | 29.56 | 7.66 | 2553 | 1408 |
| SSIM-based model | **0.95** | 0.04 | **0.01** | 0.02 | **3** | 2 |

### 3.2. Testing the SSIM-Based Model

Reconstruction accuracy was examined in both time and geographical space. Figure 5 displays the mean spatial distribution of the error metrics: SSIM, MSE, and RE, respectively. Since both SSIM and MSE output a single value for every $16 \times 16$ point area, the single value was used to quantise the entire respective area. The RE returns the appropriate amount of values—hence the higher resolution. From Figure 5, it is apparent that the highest reconstruction accuracy is achieved at the open sea areas. The accuracy diminishes as the distance from the coast decreases—the gradient is higher towards the west coast than the east coast. The lowest accuracy is contained in the north-west area of the Adriatic by the mouth of the Po River. Figure 6 examines the metrics as a function of time. Intra-annually, RE fluctuates slightly, achieving its minimum during late spring. SSIM and MSE seem to share similar behaviour—higher accuracy is achieved during summer and the lowest

accuracy occurs during late autumn. Interannually, SSIM, MSE, and RE are generally stable—with the caveat of a sudden drop in accuracy during 2013 and 2014.

These errors prompted a secondary investigation in order to determine the underlying cause of the diminished reconstruction accuracies in 2013 and 2014. This included an examination of the mean values of $chl_a$ contained in the sets used for training and testing the model. The mean value and standard deviation of $chl_a$ of both sets is $0.40 \pm 0.72$ mg m$^{-3}$. Instances where the MSE of reconstruction was greater than $0.40$ mg$^2$ m$^{-6}$ numbered 151 samples in total. The mean value and standard deviation of $chl_a$ of the aforementioned samples was $4.88 \pm 1.76$ mg m$^{-3}$. Furthermore—these samples were all localised at the mouth of River Po. The examination of $chl_a$ throughout the years indicated a sudden increase in the aforementioned area in the years 2013 and 2014—coinciding with the drop in accuracy in Figure 6.

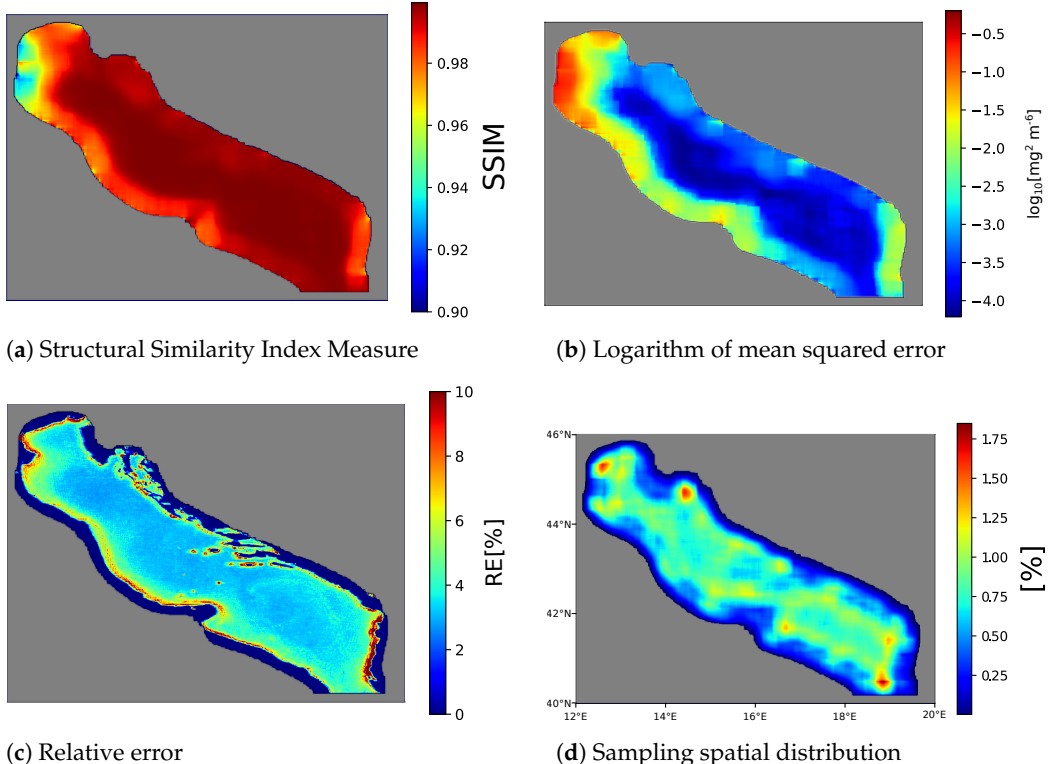

(**a**) Structural Similarity Index Measure

(**b**) Logarithm of mean squared error

(**c**) Relative error

(**d**) Sampling spatial distribution

**Figure 5.** Geospatial distribution of the three error metrics—Structural Similarity Index Measure (SSIM) in (**a**), mean squared error (MSE) in (**b**), relative error (RE) in (**c**). (**d**) displays the spatial distribution of test data sampling.

The eight most representative vectors from the dataset were selected based on the least vector norm given the eight patterns determined by the GNG from Section 2.5 (Figure 7). These vectors were masked according to the procedure and fed into the SSIM-based CCGAN to reconstruct the missing parts of the data. The visualisation of the results is displayed in Figure 7. CCGAN successfully managed to reconstruct all the closest possible data to characteristic patterns. Generally, the patterns describe certain distributions of $chl_a$, temperature and bathymetry. Pattern E represents a homogeneous distribution of low $chl_a$, typical for the southern parts of the Adriatic. High $chl_a$ attributed to the Po river is displayed in pattern D. Eddy-like distributions can be seen in patterns A and G, the former represents a positive and the latter a negative eddy. All patterns are subject to some smoothing effects—but features (eddies, local minimums and maximums, small-scale features) still prevail. The temperature in the Adriatic Sea oscillates between the summer maximum—303 K—and the winter minimum—285 K. This range has not been encapsulated by the characteristic patterns—all of the patterns oscillate around the median value. The $chl_a$ patterns that triggered the BMUs occurred during both

summer and winter months, which resulted in the averaging of the temperature. Temperature has no clear influence on the reconstruction accuracy. Bathymetry, on the other hand, has been shown to be associated with $chl_a$, which is to be expected as bathymetry is connected with physical processes that influence $chl_a$—river discharge, upwelling, coastal processes, etc.

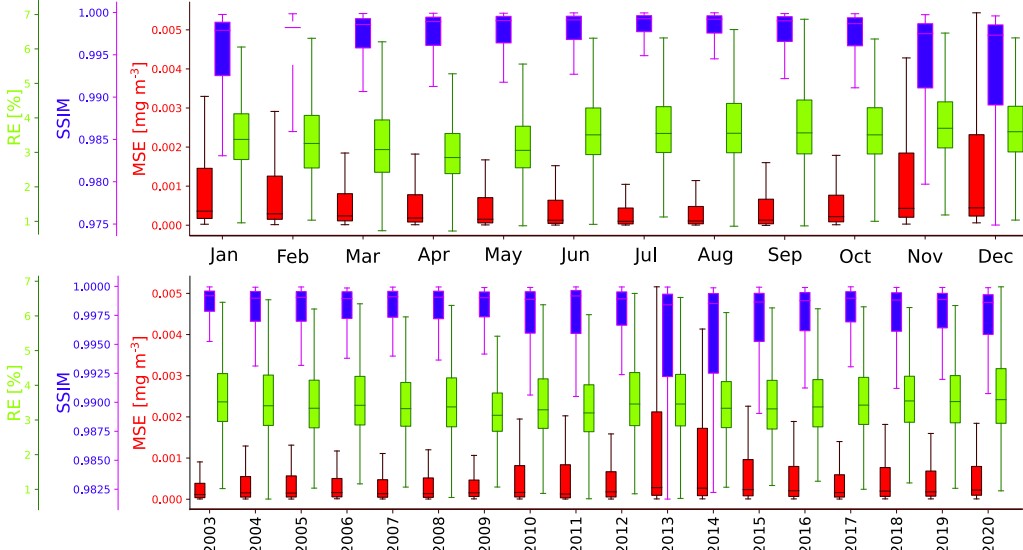

**Figure 6.** Intra-annual and interannual distributions of Structural Similarity Index Measure (SSIM, blue), mean squared error (MSE, red), and relative error (RE, green). Top part displays the monthly dependency, while the bottom displays the yearly. Boxplots display the minimum, maximum, mean, lower, and upper quartile of each metric.

Another examination evaluated the effects of including various combinations of proxy variables. To date, each data matrix contained $chl_a$, SST and bathymetry data. Three additional models were trained. Considering how $chl_a$ is crucial for reconstruction, $chl_a$ was kept in all models, while SST and/or bathymetry were excluded. Based on the possible combinations, training and testing dataset matrices were modified, by removing the appropriate data, and models were trained on the newly obtained sets. Reconstruction accuracies were compared to the model that used all three types of data. Naturally, each model was tested using only the appropriate matrices. Based on the results, the $chl_a$ and SST and bathymetry model performed the best. Removing either the SST or the bathymetry data decreases the reconstruction accuracy. It would seem that removing SST affects the SSIM and RE accuracy more than the removal of bathymetry, but the mean MSE benefits from retaining SST in favour of bathymetry. Removing both the SST and bathymetry improves the SSIM and RE accuracy compared to the $chl_a$ and SST and the $chl_a$ and bathymetry models, but degrades the mean MSE score when compared to $chl_a$ and SST model. Therefore, it could be concluded that SST and bathymetry provide complementary data which improve the reconstruction accuracy, but the inclusion of just one or the other has negative effects on the reconstruction, at least when considering the sale of the entire Adriatic.

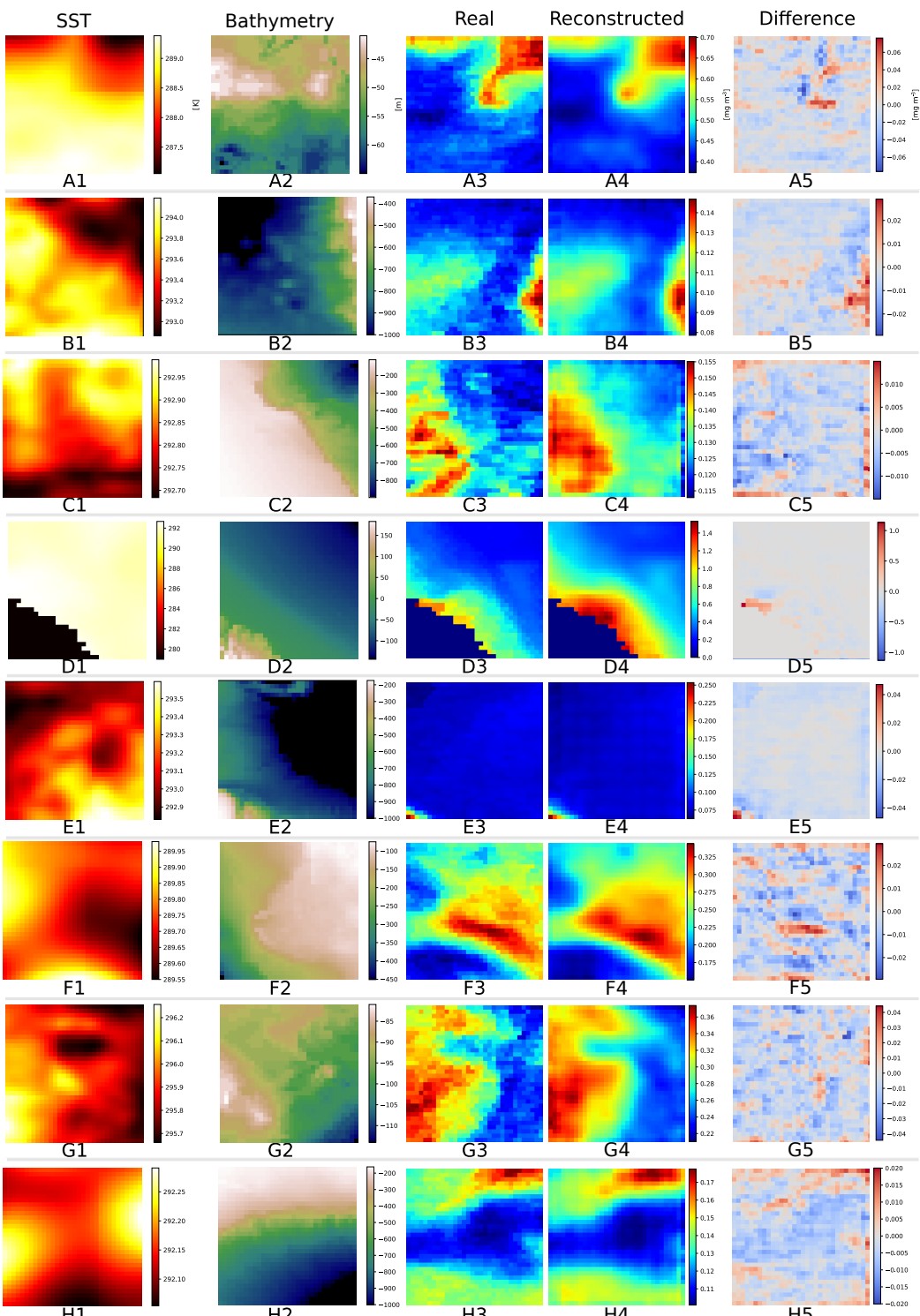

**Figure 7.** Reconstruction of the masked parts of data matrices as selected by the least vector norm based on the characteristic patterns (**A1**–**A5**,**B1**–**B5**,**C1**–**C5**,**D1**–**D5**,**E1**–**E5**,**F1**–**F5**,**G1**–**G5**, and **H1**–**H5**) derived by double Growing Neural Gas. Columns 1 and 2 display the proxy variables—sea surface temperature (SST) and bathymetrical data, in that respective order. Column 3 is the masked part of the real data chlorophyll *a* concentration data presented to the CCGAN algorithm, column 4 displays the CCGAN's reconstructed output and column 5 contains the respective difference between the target and reconstructed data.

## 4. Conclusions

This paper described a method using deep neural network for chlorophyll *a* concentration data reconstruction. The method uses Context Conditional Generative Adversarial Network architecture which was trained on satellite-obtained data. As the most important change in the canonical CCGAN implementation, the modification of the optimisation function was pointed out, along with the introduction of the Structural Similarity Index Measure as a measure of (dis-)similarity between data matrices. Such a modification improved the convergence properties and provided better overall reconstruction results. The final results of the average error of 0.06 mg m$^{-3}$ and relative error of 3.87% demonstrate outstanding performance. This is especially visible when compared to the required measurement uncertainty of essential climate variables of 30% [26]. When compared to the ISO standard [25] for laboratory measurements of chlorophyll, one can notice that the results are quite competitive, since the standard declares the coefficient of variation to be 4.3%.

CCGAN successfully managed to reconstruct small-scale features (a few kilometres in size) in the distribution of the chlorophyll *a* concentration. Despite the smoothing effects, the physical credibility of the reconstruction is preserved. Differences between the target data and the reconstruction output are randomised—there is no clear bias towards overestimation or underestimation. A slight seasonal error is present—the error is mostly related to the seasonal distribution of chlorophyll *a* in the Adriatic. This issue could potentially be rectified by introducing additional proxy variables, for example, nutrients. CCGAN's accuracy remained stable throughout the years—with the exception of the years 2013 and 2014, when it failed to model the unusual patterns at the mouth of river Po. The highest relative errors are localised in the coastal areas—apart from the error by the Italian coast caused by the discharge of River Po, the largest errors are localised by the Albanian coast. Probable causes of this error is the exchange between the Adriatic and the Ionian Sea as dictated by the BIOS oscillating system [43] and increased coastal environmental pollution. Reconstructions in coastal areas are generally difficult due to increase, with highly localised variation dictated by anthropogenic influence (sewage discharge, industrial waste, shellfish farms), river discharge, local air–sea interaction, etc.

Apart from the description of the reconstruction method, the description of the method for combining various data into layers and the representative dataset creation was provided. It was shown that the parameters used for dataset creation generally somewhat influence the results of the reconstruction at the cost of the computation time and memory storage. It is known that the chlorophyll *a* concentration and sea surface temperature are not highly correlated [43]. Likewise, it is known that the chlorophyll *a* concentration is correlated to river discharges (i.e., Po river [44]). Furthermore, chlorophyll *a* concentration is not directly related to bathymetry, but is related to processes that are related to bathymetry (i.e., upwelling). By selecting non-highly correlated proxy variables, it was shown that CCGAN was able to utilise the non-linear correlation to improve the reconstruction accuracy. The inclusion of temperature aimed to describe the quasi-annual variability of the chlorophyll *a* concentration, while bathymetry was included to emphasise the physical processes that affect the concentration.

As a closing remark, while this paper showcased the feasibility of using GAN-based methods for missing chl$_a$ reconstruction with relatively high reconstruction accuracies, a lot of room for improvement and further testing has been left, such as minimising the noise during training or varying the training parameters [45], inclusion of additional proxy variables, splitting models based on temporal and/or geographical differences, different methods for obtaining the low resolution, and so on. Regardless, based on the practical accuracy requirements of 5–11% [25] to 30% [26] when compared to the relative error obtained by this method as described in the previous chapter, the claim that solid reconstruction accuracy can be achieved by using CCGAN holds.

**Author Contributions:** Conceptualisation, L.Ć, H.K., F.M. and S.S. formal analysis, L.Ć. and F.M. investigation, L.Ć., F.M., T.Ž., J.D. and T.J. writing—original draft preparation, L.Ć. writing—review and editing, S.S., F.M. and H.K. visualisation, L.Ć. and F.M. supervision, S.S., F.M. and H.K. project administration, H.K. and S.S. funding acquisition, H.K., S.S. and F.M. All authors have read and agreed to the published version of the manuscript.

**Funding:** This work was supported in part by Croatian Science Foundation (HRZZ) under the projects UIP-2019-04-1737, IP-2019-04-5875 StVar-Adri, and in part by the Simons Foundation Project "Collaboration on Computational Biogeochemical Modeling of Marine Ecosystems" (CBIOMES) (549947,SS).

**Institutional Review Board Statement:** Not applicable.

**Informed Consent Statement:** Not applicable.

**Data Availability Statement:** Remote sensing data used for the creation of the datasets analysed in this paper are public domain and freely available. Chlorophyll *a* concentration data were obtained from www.oceancolour.org (accessed on 1 January 2020), sea surface temperature data was retrieved from www.podaac.jpl.nasa.gov/dataset/MUR-JPL-L4-GLOB-v4.1 (accessed on 1 January 2020) and bathymetrical data are available at www.gebco.net (accessed on 1 January 2020).

**Acknowledgments:** The authors would like to extend their gratitude to Jadranka Šepić for providing additional funding via the Croatian Science Foundation (HRZZ) under the project IP-2019-04-5875 StVar-Adri, and to Shubha Sathyendranath for all the provided support.

**Conflicts of Interest:** The authors declare no conflict of interest.

## Abbreviations

The following abbreviations are used in this manuscript:

| | |
|---|---|
| ECV | Essential Climate Variables |
| GAN | Generative Adversarial Network |
| CCGAN | Context Conditional Generative Adversarial Network |
| ISO | International Organization for Standardization |
| OC-CCI | Ocean Colour Climate Change Initiative |
| SST | Sea Surface Temperature |
| chl$_a$ | Chlorophyll *a* Concentration |
| MSE | Mean Squared Error |
| SSIM | Structural Similarity Index Measure |
| RE | Relative Error |
| WR | Window Resampling |
| RA | Resample Allowed |
| LA | Land Allowed |
| BMU | Best Matching Unit |

## Appendix A. Dataset Sampling

The ground requirement to train and evaluate the model was to keep the dataset completely gap-free. Therefore, $64 \times 64$ areas that contain any missing data due to obnubilation (from clouds, rain, sunglint) or due to quality control were not subject to sampling. Based on the entire time span, generally, open seas data points are mostly available. However, points in a thin zone (less than 5 km) along the coastline are available around 50% of the time, and in some areas, even less than 20%. The east part of the Adriatic, because of the Adriatic archipelago, is more affected by the optical complexity of the coastal waters than the west part. This is due to the increased optical complexity of coastal waters [46,47], which significantly hinders the retrieval of chl$_a$ [48,49]. If a fixed, predefined geographical land–sea mask is used, it would result in overwhelmingly numerous samples of the open sea and few, if any, samples of coastal areas. To avoid this, a much looser definition of land–sea mask was used. Opted for was a dynamic land–sea mask, depending on the availability of the data in the coastal area. Figure A1 depicts this, where Figure A1a shows the geographical land–sea mask, Figure A1b shows the dilated land–sea mask, where dilation

was performed based on the availability of the data, and Figure A1c shows the difference between the two. Rather than obtaining the dilated version of the land–sea mask by using morphological transformations, it was obtained by defining areas which contain data less than 40% of time as land, while the rest was defined as sea. Obviously, this degraded the geographical accuracy on a global scale; however, the $64 \times 64$ areas only contain local information, therefore by doing this preservation of the most important information the distribution of data and land and sea areas was achieved.

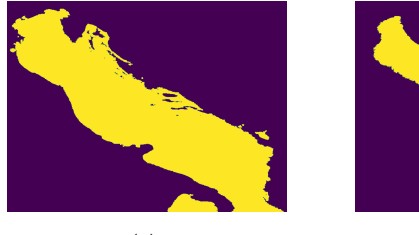
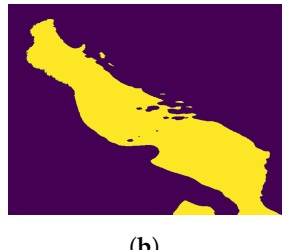
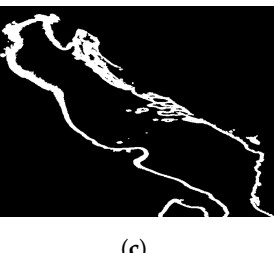

| (a) | (b) | (c) |

**Figure A1.** Dilated land–sea mask: (**a**) geographical land–sea mask; (**b**) the dilated land–sea mask; and (**c**) the difference. Land points are purple, sea points are yellow. White represents the geographically accurate sea points that were classified as land points using the dilated land–sea mask.

Unlike gaps in data in open seas, which are mostly caused by cloud coverage, gaps in the coastal zone are usually caused by $chl_a$ deriving the algorithm's inability to deal with optically complex waters [48,49]. This issue surfacing from the algorithm itself could easily be circumvented by just applying the dilated land–sea mask. However, on some occasions, $chl_a$ was obtained in the white areas in Figure A1c. In order not to discard the data when present during the sampling, the algorithm took into consideration points in the white areas if they were available, otherwise, it considered it to be land rather than missing data. This is the reasoning behind the "dynamic" adjective in the dynamic land–sea mask.

After applying the dynamic land–sea mask, which allowed for near-shore sampling, the goal was to create a dataset that would be as continuous as possible, while being as spatially and temporally representative as possible. In order to do so, several heuristic rules were put in place. With the dynamic land–sea mask being used, sampling could now result in numerous samples containing an unnecessarily large percentage of land points as opposed to sea points. This would increase significantly the size of the dataset, but would either have no positive or a negative effect on the spatial representativeness of the set. Simplified, the algorithm for sampling needed to be barred from extracting $64 \times 64$ samples (containing 4096 points in total) that contain too few sea points (e.g., one sea point and 4095 land points). To do so, a *Land Allowed* (LA) parameter rule was enforced, which verified that each candidate sample contains a percentage of land points no greater than the one given by the parameter. Furthermore, $chl_a$ exhibits a certain degree of seasonality in the Adriatic. This seasonality is in tune with the climatology of the Adriatic: spring and summer skies are generally cloud-free, while autumn and winter months display significantly larger and more frequent cloud cover. This in turn results in more missing points during the autumn–winter period than in the spring–summer period. If this difference is not regulated, the created dataset could contain unproportionally more spring–summer samples, skewing the temporal distribution, which may result in autumn–winter examples being reconstructed with the incorrect seasonality. Therefore, the second rule limits the number of times an area could be sampled in a day, based on the average number of times points in the area have been sampled for that date. The parameter of this rule is named *resample allowed* (RA). This parameter is tied to the *counter* parameter, which, as the name suggests, counts the number of times a point has been sampled. This *counter* is updated after each successful sampling by a value called resample-increment, denoted by $N$. Surface $chl_a$ distribution is a complex system. While the $64 \times 64$ sample should encapsulate most oceanographic features (such as fronts, eddies, etc.), the surrounding area still has an influence on such features. Having a clear-cut separation between two

neighbouring samples would most likely result in unnatural boundary artefacts [50,51] when attempting to reconstruct the missing data. To reduce this error, a continuous dataset is required. To achieve this, overlapping in sampling is allowed, but is governed by the third rule which dictates how many points around the centre of the sampled area will be tagged as already sampled after each $64 \times 64$ sample. The width and height of this square window are defined by the *Window Resampling* (WR) parameter. Points within this WR window obtain their sampling counter updated by value $N$ after each sampling. A toy example of an algorithm on a downsampled matrix is displayed in Figure A2.

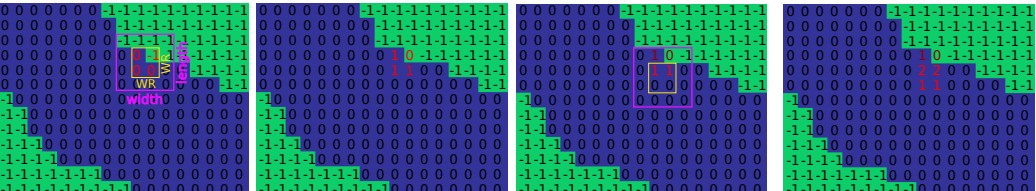

**Figure A2.** Depiction of how the algorithm used for data sampling updates the counter variable to assure the representativeness of the set. Green pixels represent land, whilst blue pixels represent sea. The black number denotes sampling the counter variable for each pixel. For details, see text.

Figure A2 depicts a toy example for a downsampled matrix. In this example, the output matrix is reduced from $64 \times 64$ points to just $4 \times 4$ points and is shown as a pink rectangle. The yellow rectangle is defined by the parameter WR, in this case, $WR = 1$. Each point has an auxiliary integer variable *counter*. Based on the dynamic land–sea mask, the *counter* of land points is assigned the value $-1$, while sea points are assigned 0. The random pair of latitude and longitude (lat, lon) is selected. After verifying that there are no missing points, the LA condition is satisfied by determining the percentage of green points inside the pink window and the RA condition is satisfied by comparing the value of RA to the mean value of the counter within the yellow square, in that specific order—the part of the matrix within the pink rectangle is sampled and added to the dataset. After sampling, only the points within the yellow square, whose dimensions are defined by the WR parameter, receive a *counter* update, by adding $N = 1$ to their value (updated *counter* values are shown as red numbers). In the next step, a new (lat, lon) pair is selected (in this example, shifted by one point to south) and the process is repeated. While the algorithm tries to find every possible (lat, lon) pair, notice that it is generally a non-deterministic algorithm, as the resulting dataset depends on the order in which the (lat, lon) pairs are selected.

**Appendix B. Sanity Tests**

Dataset creation itself is a tedious and computationally intensive process that might introduce significant biases into the machine learning model. In order to avoid any of these, the statistical properties of the dataset, as well as the effects that parameters used for dataset creation may have on the model output were investigated. As explained earlier, the properties of the dataset are governed by the parameters WR, RA, and LA. Effects each parameter has on the dataset have been tested by varying its value of it, while keeping the values of the other two parameters constant. Starting with the dataset created with WR = 16, RA = 2.5, and LA = 0.50, six additional datasets were created, by varying WR from 8 to 32, RA from 1.5 to 3.5, and finally by changing LA from 0.25 to 0.75. To select the best possible combination of the parameters, seven separate neural networks were trained on their respective datasets and their reconstruction accuracies were compared. While this approach might reveal which dataset gives the most accurate training, it does not quite verify the representativeness of the dataset compared to the initially available data. Therefore, three sanity tests were performed on the WR = 16, RA = 2.5, LA = 0.50 dataset in order to approximate and validate the spatial and temporal distribution along with the numerical representativeness. Numerical representation was also tested for the six remaining datasets. The results of the numerical representativeness are displayed in

Table A1 and the results of spatial and temporal tests of the WR = 16, RA = 2.5, LA = 0.50 dataset are displayed in Figure 5. All datasets performed similarly, with the general trend that bigger datasets capture the numerical distribution more closely.

**Table A1.** Numerical representativeness of the datasets with regard to the mean $\text{chl}_a$. While it may seem that the original *.nc* files contain a significantly higher mean $\text{chl}_a$ value, it is important to realise that natural obnubilation does not follow the rectangular shape of the matrices, so some deviation is to be expected.

| WR | RA | LA | $\mu_{\text{chl}_a}$ (mg m$^{-3}$) | $\sigma_{\text{chl}_a}$ (mg m$^{-3}$) | Number of Training Matrices |
|---|---|---|---|---|---|
| 8 | 2.5 | 0.50 | 0.4254 | 0.7603 | 2,561,580 |
| 16 | 1.5 | 0.50 | 0.4147 | 0.7404 | 418,260 |
| 16 | 2.5 | 0.25 | 0.4085 | 0.7168 | 579,360 |
| 16 | 2.5 | 0.50 | 0.4189 | 0.7485 | 669,560 |
| 16 | 2.5 | 0.75 | 0.4306 | 0.7706 | 789,760 |
| 16 | 3.5 | 0.50 | 0.4215 | 0.7538 | 920,480 |
| 32 | 2.5 | 0.50 | 0.4048 | 0.7214 | **252,100** |
| | | *.nc* files | 0.4831 | 1.005 | |

Once datasets were extracted, all data were normalised to the range $[0, 1]$ to improve the training stability. The normalisation was performed on all three channels. Bathymetry was normalised in accordance with the cutoff values that were assigned previously. $\text{chl}_a$ and SST were examined for the highest available values. The lowest value of both variables is zero, as defined by the value assigned to the land points. SST's highest recorded value was 303.57 K. $\text{chl}_a$ normalisation required more work. Namely, due to landlocked bodies of water, such as Valli di Comacchio, the maximum $\text{chl}_a$ was around 99 mg m$^{-3}$. This value is too high for the purpose of reconstructing sea $\text{chl}_a$, so a cutoff needed to be made. Examining the maximum values of $\text{chl}_a$ per matrix, it was determined that less than 0.5% of matrices had a maximum value of $\text{chl}_a$ greater than 20 mg m$^{-3}$, and around 3% had a maximum value of $\text{chl}_a$ greater than 10 mg m$^{-3}$. The cause of such unusually high $\text{chl}_a$ is the discharge of Po river [52]. Since this paper is dealing with the reconstruction of $\text{chl}_a$ on the entire Adriatic, allowing such high values into the dataset could potentially skew the data distribution in an unfavourable way, which could diminish the reconstruction accuracy. Therefore, a normalisation cutoff was set at 12 mg m$^{-3}$, so that the dataset would not be affected by unreasonably high $\text{chl}_a$. Reconstruction efforts were tested on previously unseen matrices from the test set. The train–test split was performed in an 80:20 ratio [53].

**Appendix C. Training and Testing the Datasets**

In order to decide the most appropriate dataset among the seven datasets described, a comparison was made of reconstruction results in terms of SSIM, MRE, and RE. The reconstruction accuracy was evaluated on the test set, in accordance with the experiment performed on the SSIM-based model. As for the effects of varying the WR, RA and LA parameter values, there are several findings to point out. Firstly, increasing the WR decreases SSIM and MRE values, while RE seems to be lowest for WR = 16. The RA value seems to be proportional to SSIM, inversely proportional to MSE, and RE seems to be lowest for RA = 2.5. Finally, increasing LA decreases both MSE and RE, while no conclusive correlation to SSIM is seen. As hinted by the results in Appendix B, the WR = 8, RA = 2.5, LA = 0.50 dataset model scored the best accuracies across all three metrics. However, the WR = 8, RA = 2.5, LA = 0.50 dataset model took significantly longer to train, around 130 h, while the next biggest dataset took around 50 h to train. Since the remaining models did not perform significantly worse, the smallest dataset (the WR = 32, RA = 2.5, LA = 0.50)

model, which took around 14 h to train, was selected as the most efficient one that was used in the remainder of the paper. All training was GPU accelerated, using four NVIDIA GeForce RTX 2080 Ti graphics cards, on a 128 GB RAM system. CPU used was AMD Ryzen Threadripper 1920X 12-Core.

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
