# Peer review of "CCGAN as a Tool for Satellite-Derived Chlorophyll a Concentration Gap Reconstruction"

_jmse, doi:10.3390/jmse11091814_

Round 1

Reviewer 1 Report

The paper is very interesting, it is creative in taking SST and bathymetry data into account when reconstruction of Chl-a remote sensing data. The method is well presented and the results are credible. The method gave new solution for gappy satellite retrieved marine parameters. I suggest the paper can be accepted with minor revision.

Comment 1: the section 4 “discussion and conclusion” should be “conclusion”, it is too short as discussion, section 3 could be “results and discussion”.

Comment 2: Give the full name of abbreviation when they first appear even they are used frequently, for example, Line 58,59: Give the full name of ISO and ECV, BMUs. The authors need to check through the whole paper. Line 77: “chlorophyll a concentration data (chla)”, give the abbreviation when it first appear in the text.

Comment 2: the English writing is casual in some paragraph, for example: Line 77-78: “To determine if a machine learning model for reconstructing missing chlorophyll a concentration data(chla)could be improved other variables have been included.” In addition, Line 3: “By adjusting the loss functions of the network to focus on the structural credibility of the reconstruction high numerical and structural reconstruction accuracies have been achieved in comparison to the original network architecture.”, the sentence lacks a comma? The authors need to check through the paper.

Reviewer 2 Report

The presented results of modifying the Context Conditional Generative Adversarial Network to fill the chlorophyll concentration gaps are of unconditional interest to the Ocean Color community. The manuscript is worth publishing.

Comments:

1. It is necessary to indicate the region of study in the annotation.

2. It is recommended to provide a geographical map of the region under study with a world map insert, which is necessary for readers who are unfamiliar with European seas.

3. It is necessary to add coordinates to Figure 3.

4. Unfortunately, I was not able to understand how the reconstructed data in Figure 5 was obtained. The text says about the restoration of the missing parts, but it is not at all clear from the content of the manuscript how these parts were masked. It is recommended to add information about masked data. The image of the masked areas will allow the reader to get a visual representation of how the algorithm works.

5. The decoding of the abbreviation MCE appears in the text after its first mention.

6. Apparently, in Figure А2 there are no numbers of its parts (1-4), which are referenced in the text.

7. The parameters of the computers used should be specified so that the network training time information given in Appendix C becomes more understandable to the reader.

Minor typos should be corrected, such as "remains table" on line 321.

What does "Wet points" mean on line 65?
